# Tensor Networks for Medical Image Classification

**Raghavendra Selvan** [1]                                        RAGHAV@DI.KU.DK
**Erik B Dam** [1]                                                ERIKDAM@DI.KU.DK
[1] *Department of Computer Science, University of Copenhagen, Denmark*

## Abstract

With the increasing adoption of machine learning tools like neural networks across several domains, interesting connections and comparisons to concepts from other domains are coming to light. In this work, we focus on the class of Tensor Networks, which has been a work horse for physicists in the last two decades to analyse quantum many-body systems. Building on the recent interest in tensor networks for machine learning, we extend the Matrix Product State tensor networks (which can be interpreted as linear classifiers operating in exponentially high dimensional spaces) to be useful in medical image analysis tasks. We focus on classification problems as a first step where we motivate the use of tensor networks and propose adaptions for 2D images using classical image domain concepts such as local orderlessness of images. With the proposed locally orderless tensor network model (LoTeNet[1]), we show that tensor networks are capable of attaining performance that is comparable to state-of-the-art deep learning methods. We evaluate the model on two publicly available medical imaging datasets and show performance improvements with fewer model hyperparameters and lesser computational resources compared to relevant baseline methods.

**Keywords:** Tensor Networks, Image classification, histopathology, thoracic CT, lesions

## 1. Introduction

Kernel methods revolutionised pattern recognition and machine learning with the class of support vector machines (SVMs) in the 90's, based on the fundamental insight that difficult problems in low dimensions may become easier when lifted to high dimensional spaces (Boser et al., 1992; Cortes and Vapnik, 1995; Hofmann et al., 2008).

An efficient approach to dealing with such high dimensional spaces can be with *tensor networks*, also known as tensor trains. Tensor networks are factorisations of high dimensional tensors into networks of low rank tensors and come with a class of efficient algorithms to perform these approximations (Oseledets, 2011; Bridgeman and Chubb, 2017). The number of parameters needed to specify an $N$ dimensional tensor using tensor networks can be drastically reduced, from exponentially increasing with $N$ to a polynomial dependence on $N$ (Perez-Garcia et al., 2006).

Recently, there has been an increasing interest in using tensor networks in the context of supervised machine learning, specifically focused on image classification tasks (Stoudenmire and Schwab, 2016; Efthymiou et al., 2019). These methods rely on transforming the 2-d input images into 1-d vectors before encoding them into high dimensional spaces. As a consequence of this flattening these methods are constrained to work with images of small spatial resolution ($12 \times 12$ px to $28 \times 28$ px), and focus on employing improved

---

1. Official repository: https://github.com/raghavian/loTeNet_pytorch/

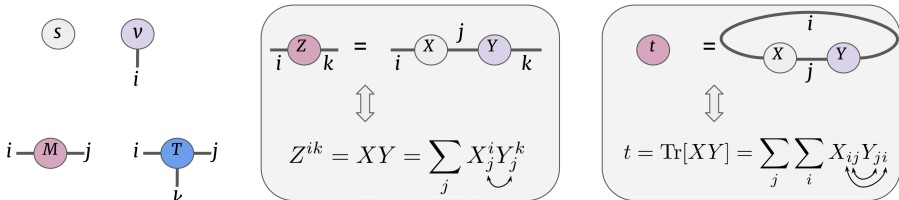

Figure 1: (left) Tensor notation depicting a scalar $s$, vector $v^i$, matrix $M^{ij}$ and a general 3-D tensor $T^{ijk}$. (center) Tensor notation for matrix multiplication or *tensor contraction*, which are used extensively in the matrix product state networks used in this work. We adhere to the convention that the contracted indices are written as subscripts. (right) Tensor notation for trace of product of two matrices.

flattening strategies to maximize the retained pixel correlation (Stoudenmire and Schwab, 2016; Efthymiou et al., 2019). For small enough images (like in MNIST or Fashion MNIST datasets) there is some residual correlation in the flattened images which can be exploited using tensor networks. In medical imaging tasks, however, images with such low spatial resolutions are rarely encountered. Further, the information lost by flattening of images in medical imaging tasks can be crucial as many decisions can be dependent on the global structure of the pixels.

In this work, we extend the use of tensor networks to be useful in classification of medical images. We propose the locally orderless tensor network or LoTeNet (pronounced "low tenet") inspired from the classical theory of locally orderless images in Koenderink and Van Doorn (1999). According to the theory of locally orderless images, statistics from small neighbourhoods in images can be derived by ignoring the local order of pixels while still capturing the global structure by operating at different scales. The proposed LoTeNet model is used to perform linear classification in high dimensional spaces and it is optimized end-to-end by backpropagating the error signal through the tensor network. We present experiments on two medical imaging datasets: PCam dataset with histopathology images (Veeling et al., 2018) and LIDC-IDRI dataset with thoracic CT images (Armato III et al., 2004). Our model fares comparably to state-of-the-art deep learning models with fewer model hyperparameters and utilizing a fraction of the GPU memory when compared to their CNN counterparts.

## 2. Background and Problem Formulation

### 2.1. Tensor Network Notations

Tensor networks and operations on them are described using an intuitive graphical notation, introduced in Penrose (1971). Figure 1 (left) shows the commonly used notations for a scalar $s$, vector $v^i$, matrix $M^{ij}$ and a general 3-D tensor $T^{ijk}$. The number of dimensions of a tensor are captured by the number of edges emanating from the nodes denoted by the edge indices. For instance, the vector $v^i$ has one dimension indicated by the edge with index $i$ and a 3-d tensor has three indices $(i, j, k)$ depicted by the three edges, and so on.

Operations on high dimensional tensors can be succinctly captured using tensor networks as shown in Figure 1 (center) where matrix product is depicted, which is also known as *tensor contraction*. The edge between the tensor nodes $X_j^i$ and $Y_j^k$ is the dimension subsumed in matrix multiplication resulting in $Z^{ik}$. More thorough introduction to tensor notations can be found in Bridgeman and Chubb (2017).

## 2.2. Linear model in high dimensions

A linear model in a sufficiently high dimensional space can be very powerful (Novikov et al., 2016). In SVMs, this is accomplished by the *implicit* mapping of the input data into an infinite dimensional space using radial basis function kernels (Hofmann et al., 2008). In this section, we describe the procedure followed in this work to map the input data into a higher dimensional space.

Consider an input vector $\mathbf{x} \in [0,1]^N$, which can be obtained from a flattened 2-d image with $N$ pixels in total and intensity values normalized in the interval $[0,1]$. A commonly used feature map for tensor networks is obtained by taking the *tensor product* of pixel wise feature maps (Stoudenmire and Schwab, 2016):

$$\Phi^{i_1,i_2,\ldots i_N}(\mathbf{x}) = \phi^{i_1}(x_1) \otimes \phi^{i_2}(x_2) \otimes \cdots \phi^{i_N}(x_N) \tag{1}$$

where the local feature map acting on pixel $x_j$ is indicated by $\phi^{i_j}(\cdot)$. The local feature map is $d$-dimensional and usually is a simple non-linear function which additionally is restricted to have unit norm so that the joint feature map in Eq. (1) also has unit norm. A widely used local feature map with $d = 2$ inspired from quantum wave function analysis is (Stoudenmire and Schwab, 2016):

$$\phi^{i_j}(x_j) = [\cos(\frac{\pi}{2}x_j), \sin(\frac{\pi}{2}x_j)]. \tag{2}$$

The dimensionality of the joint feature map $\Phi(\mathbf{x})$ is $d^N$ due to tensor products in Eq. (1), as the local feature maps in Eq. (2) are of dimensionality $d = 2$. The joint feature map $\Phi(\mathbf{x})$ basically maps each image as a vector in the $d^N$ dimensional feature space. For an RGB image, or other image modalities with $C$ input channels as commonly encountered in medical imaging, the local feature map can be applied to each channel separately such that the resulting space is of dimension $(d \cdot C)^N$ (Efthymiou et al., 2019).

Given the high dimensional feature map $\Phi(\mathbf{x})$ of Eq. (1) for the input data $\mathbf{x}$, a decision rule for a multi-class classification task can be formulated of the form:

$$f(\mathbf{x}) = \arg\max_m f^m(\mathbf{x}), \tag{3}$$

where $m = [0, 1, \ldots M - 1]$ are the $M$ classes,

$$f^m(\mathbf{x}) = W^m \cdot \Phi(\mathbf{x}). \tag{4}$$

and the weight tensor $W^m$ is an $N + 1$ dimensional tensor with output tensor index $m$.

In tensor notation, the linear model of Eq. (4) is depicted in Figure 2 (Step 1) where the first column of gray nodes are the individual pixel feature maps of feature dimension $d$. The pixel feature maps are connected to the weight tensor $W^m$ along $N$ edges and $W^m$ has one output dimension marked with index $m$.

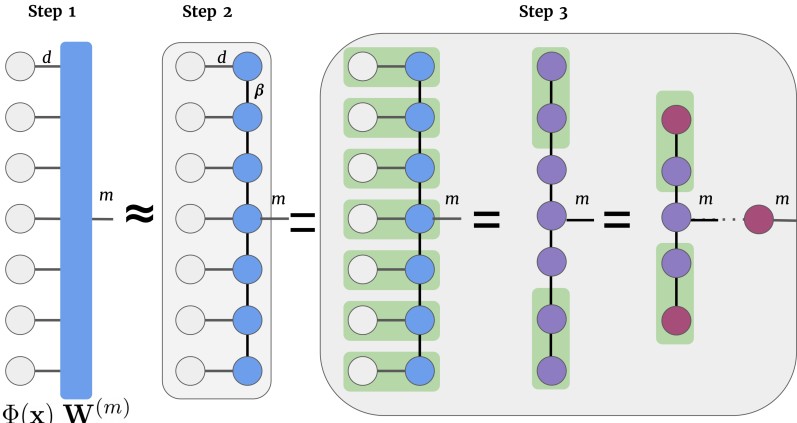

Figure 2: (Step 1) Linear model of Eq. (4) in tensor notation. (Step 2) MPS approximation of the linear model. (Step 3) Series of tensor contractions done with MPS to compute $W^m \cdot \Phi(\mathbf{x})$ in Eq. (4)

The $N+1$ dimensional weight tensor $W^m$ results in total of $M \cdot d^N$ number of weights. Even for a relatively small gray scale image, say of size $100 \times 100$, the total number of components in $W^m$ can be massive: $2 \cdot 2^{10000} \approx 10^{3010}$. In the next section we will see how tensor networks can represent such high dimensional tensors with parameters that grow linearly with $N$ instead of growing exponentially with $N$.

### 2.3. Matrix Product State (MPS)

Consider two 1-d vectors, $v^i$ and $u^j$ with dimension indices $i$ and $j$ respectively. The tensor (outer) product of these two vectors yields a 2-d matrix $X^{ij}$. The matrix product state (MPS) Perez-Garcia et al. (2006) is a type of tensor network that expands on this notion allowing the factorization of an $N$-dimensional tensor (with $N$ edge indices) into a chain of rank-3 tensors (with three edge indices) except on the borders where they are of rank-2, as shown in Figure 2 (Step 2). Concretely, a tensor of $N$ dimensions with indices $i_i, i_2, \ldots i_N$ can be approximated using lower rank tensors $A^{i_j}$ as

$$W^{m,i_1,i_2,\ldots i_N} = \sum_{\alpha_1,\alpha_2,\ldots \alpha_N} A^{i_1}_{\alpha_1} A^{i_2}_{\alpha_1\alpha_2} A^{i_3}_{\alpha_2\alpha_3} \ldots A^{m,i_j}_{\alpha_j\alpha_{j+1}} \ldots A^{i_N}_{\alpha_N}. \tag{5}$$

The subscript indices $\alpha_j$ are the virtual indices that are contracted and are of dimension $\beta$ which is known the *bond dimension*. The components of the intermediate lower rank tensors $A^{i_j}$ are the parameters of the MPS approximation. The placement of the output dimension $m$ on $A^{i_j}$ in Eq. (5) is an arbitrary choice and can be adapted during the optimisation (Stoudenmire and Schwab, 2016). Note that any $N$ dimensional tensor can be represented exactly using an MPS if $\beta = d^{N/2}$, where $d$ is the feature dimension. In most applications, however, $\beta$ is fixed to a small value or allowed to adapt dynamically when the MPS is used to approximate a high dimensional tensor (Perez-Garcia et al., 2006; Miller, 2019).

The decision function in Eq. (4) can now be computed using the MPS approximation of $W^m$ in Eq. (5) depicted in Figure 2 (Step 2). The order in which the tensor contractions are performed can yield a computationally efficient algorithm. The original MPS algorithm (Perez-Garcia et al., 2006) starts from one of the ends, contracts a pair of tensors to obtain a new tensor which is then contracted with the next tensor and this process is repeated until the output tensor is reached. The cost of this algorithm as $N \cdot \beta^3 \cdot d$ when compared to the cost that scales as $d^N$ without the MPS approximation. In this work we use the MPS implementation in Miller (2019) which contracts the horizontal edges parallely and proceeds to contract these contracted tensors vertically as depicted in Figure 2 (Step 3) and yields improved approximations (Efthymiou et al., 2019).

## 3. Methods

Recently proposed Tensor Network models for image classification purposes flatten entire 2-d images into 1-d vectors with different raveling strategies Han et al. (2018); Efthymiou et al. (2019). In contrast to these methods, we only flatten small regions of the images which can be assumed to be locally orderless and derive useful features using MPS operations Koenderink and Van Doorn (1999). We process these locally orderless regions using a hierarchical MPS tensor network which we call the Locally Orderless Tensor Network or LoTeNet, which is shown in Figure 4. This enables our model to handle larger images without losing their global structure.

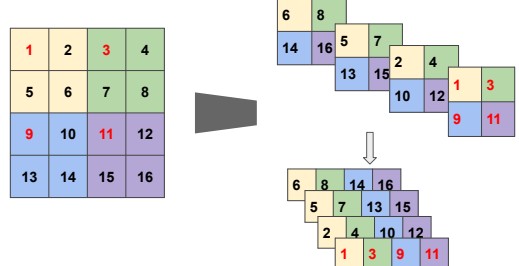

Figure 3: Squeeze operation with stride $k = 2$ which reshapes a $4 \times 4 \times 1$ image patch into $2 \times 2 \times 4$ stack. Raveling the squeezed image yields a vector of size 4 with feature dimension d=4.

### 3.1. Locally orderless Tensor Network (LoTeNet)

The locally orderless image regions are created in two steps. First, the *squeeze* operation illustrated in Figure 3 is applied on $k^2 \times k^2$ image patches where $k$ is the stride of the square kernel. This operation rearranges pixels in spatially local regions and stacks them along the feature dimension. Similar strategies have also been used in normalizing flow models to provide spatial context via feature dimensions such as in Dinh et al. (2017). The stride of the kernel $k$ decides the extent of reduction in spatial dimensions and the corresponding increase in feature dimension of the squeezed image. In the second step, the squeezed image with an inflated feature dimension of $C \cdot k^2$ is flattened from 2-d to 1-d. Flattening these local regions with spatial information along the feature axis provides our model with additional structural information. Further, the increase in $d$ makes the tensor network more flexible as it increases the dimensionality of the feature space (Stoudenmire and Schwab, 2016).

Consider the input image to the first MPS layer in Figure 4 with grids marking the different $k^2 \times k^2$ patches. Each of these patches are squeezed and input into an MPS block.

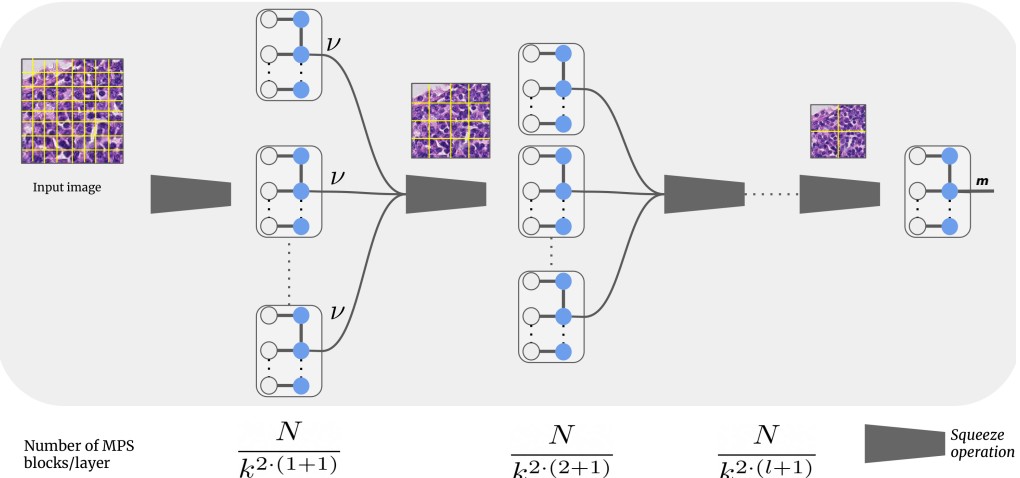

Figure 4: The proposed Locally orderless Tensor Network shown as a series of layers. Each layer consists of several MPS blocks. The squeeze operation is as described in Figure 3. The final MPS block outputs the predictions for the $M$ classes depicted as the edge with index $m$.

The MPS block first embeds these $C \cdot k^2$ vectors into the joint feature space of dimensionality $d^N$ according to Eq. (2) and Eq. (1), with $d = (2 \cdot C \cdot k^2)$. Then, the image patches in the high dimensional feature space are contracted to output a vector with dimension $\nu$. In our model $\nu$ is set to be the same as the bond dimension $\beta$. The functionality of the MPS blocks can be interpreted as summarising the patch with a vector of size $\nu$ using a linear model in the high dimensional feature space.

The output vectors from all the MPS blocks at a given layer are reshaped back into the 2-d image space. However, due to the MPS contractions in the first layer, the intermediate image map will be of lower resolution as indicated by the smaller image with fewer patches in Figure 4. This is analogous to obtaining an average pooled version of the intermediate feature maps in CNN operations. The smaller 2-d patches formed after the first layer of MPS blocks are further squeezed and contracted in the next layer of the model. This process is continued for $L$ layers and the final MPS block performs the decision contraction of Eq. (4).

### 3.2. Model Optimization

The components of the weight matrix $W^m$ (parameters of the model) are approximated using the layers of MPS blocks as described in Section 3.1. We view the sequence of MPS contractions in successive layers of our model as the forward pass and rely on automatic differentiation in PyTorch (Paszke et al., 2019) to compute the backward computation graph (Efthymiou et al., 2019). The Torch MPS module (Miller, 2019) is used to learn the MPS parameters from the training data in an end-to-end fashion.

Similar to Efthymiou et al. (2019) we minimize the cross-entropy loss between the true label $y_i \in [0, \dots, M-1]$ for each image $\mathbf{x}_i$ and the predicted label $f^{(y_i)}(\mathbf{x}_i)$ in the training



Figure 5: Four sample patches from the PCam dataset. The green region of size $32 \times 32$ px in the second and third patches denotes the presence of tumour with a positive label whereas the other two have negative labels.

set $\mathcal{D}$:

$$\mathcal{L}(f^{(y_i)}(\mathbf{x}_i)) = -\sum_{(\mathbf{x}_i, y_i) \in \mathcal{D}} \log \frac{\exp f^{(y_i)}(\mathbf{x}_i)}{\sum_{m=0}^{M-1} \exp f^{(m)}(\mathbf{x}_i)} = -\sum_{(\mathbf{x}_i, y_i) \in \mathcal{D}} \log \left( \sigma(f^{(y_i)}(\mathbf{x}_i)) \right) \qquad (6)$$

where $\sigma(\cdot)$ is the softmax operation used to obtain normalized scores that can be interpreted as the predicted class probabilities.

## 4. Data and Experiments

### 4.1. Data

We perform experiments on two publicly available datasets with the task formulated as binary classification for: metastasis detection from histopathologic scans; and detection of nodules in thoracic computed tomography (CT) scans.

**PCam Dataset**: The PatchCamelyon (PCam) dataset is a recently introduced binary image classification dataset in Veeling et al. (2018). Image patches of size $96 \times 96$ px are extracted from the Camelyon16 Challenge dataset (Bejnordi et al., 2017). A positive label indicates the presence of at least one pixel of tumour tissue in the central $32 \times 32$ px region and a negative label indicates the absence of tumour, as shown in Figure 5. Patch extraction is performed to ensure the class balance is close to $50 - 50$. We use the modified PCam dataset from the Kaggle challenge[2] which excludes duplicate image patches and consists of about $220,000$ patches for training which is further split into $80:20$ for training and validation purposes. An independent test set of about $57,500$ patches is provided for evaluating the models. All image planes are normalized to have mean and standard deviation of 0.5. We use random rotation, random horizontal and vertical flips on the training data for data augmentation.

**LIDC Dataset**: The LIDC-IDRI datatset comprises of 1018 thoracic CT images with lesions annotated by four radiologists (Armato III et al., 2004). Similar to Kohl et al. (2018); Baumgartner et al. (2019) we extract $128 \times 128$ px image patches centered on a lesion and use the preprocessed data from Knegt (2018) (shown in Figure 6 in Appendix). This yields a total of about $15,000$ patches. To transform this into a classification task we pose it as a task of predicting the presence or absence of lesions based on the four annotations. We indicate a patch to have a lesion if more than two (i.e. $\geq 3$) radiologists have annotated presence of a lesion and a negative label in the remaining two cases. The

---

2. https://www.kaggle.com/c/histopathologic-cancer-detection

Table 1: Performance comparison on PCam dataset (left) and LIDC dataset (right). For the LIDC models we also compare the GPU memory utilisation shown in gigabytes.

| PCam Models | GPU (GB) | AUC | LIDC Models | GPU (GB) | AUC |
|---|---|---|---|---|---|
| Rotation Eq-CNN | 11.0 | 0.963 | LoTeNet (ours) | 0.7 | 0.874 |
| DenseNet | 10.5 | 0.962 | Tensor Net-X ($\beta = 10$) | 4.5 | 0.847 |
| LoTeNet (ours) | 0.8 | 0.943 | DenseNet | 10.5 | 0.829 |
| Tensor Net-X ($\beta = 10$) | 5.2 | 0.908 | Tensor Net-X ($\beta = 5$) | 1.5 | 0.823 |

binary task then transforms it into capturing the majority vote amongst the radiologists. We split the dataset into $60 : 20 : 20$ splits for training, validation and a hold-out test set.

### 4.2. Experiments and Results

The proposed model (LoTeNet) is evaluated with $L = 3$ layers and a kernel size $k = 2$ (for the squeeze operation) and it is implemented based on the efficient MPS implementations in Miller (2019). The only critical hyperparameter inherent to LoTeNet is its bond dimension $\beta$; it was set to $\beta = 5$ obtained from the range $[2, 4 \ldots 20]$ based on the performance on the PCam validation set (Figure 8 in Appendix). We used the Adam optimizer (Kingma and Ba, 2014) with a learning rate of $5 \times 10^{-4}$ and a batch size of 512. Models were assumed to have converged if there was no improvement in validation AUC over 5 consecutive epochs and the model with the best validation performance was used to predict on the test set. All experiments were run on a single Tesla K80 GPU with 12 GB memory. The same settings were used for the experiments on LIDC dataset.

We compare performance of our model with DenseNet baseline with 4 layers and a growth rate of 12 as described in Huang et al. (2017) and also the single layer MPS model in Efthymiou et al. (2019) reported as Tensor Net-X. Additionally, we compare the PCam dataset performance to the rotation equivariant CNNs method which also introduced the dataset (Veeling et al., 2018). We report area under the ROC curve (AUC) as the metric to compare the different models.

The test set performance for PCam and LIDC datasets with the relevant comparing methods are are reported in Table 1. We notice that LoTeNet attains an AUC of 0.943 on the PCam dataset which is comparable with the methods of Veeling et al. (2018) and Dinh et al. (2017). There is a clearer improvement when compared to DenseNet on the LIDC dataset. Further, LoTeNet outperforms the Tensor Net-X with a single layer MPS (Efthymiou et al., 2019) on both datasets. LoTeNet takes about 3 minutes per training and validation epoch on the PCam dataset and $30s$ on the LIDC dataset.

## 5. Discussion and Conclusions

The most important hyperparameter of any MPS model is its bond dimension $\beta$ as it controls the quality of approximation of the high dimensional tensor. In our proposed model, LoTeNet, which is composed of layers with MPS blocks, we noticed the performance to be robust to the changes in $\beta$ (Figure 8 in Appendix). This is consistent with other findings where bond dimension after a certain number (around 10) has shown to have no

impact on the performance of the models (Efthymiou et al., 2019). Due to the distributed nature of approximation in LoTeNet across several layers this is all the more pronounced and we find only minor fluctuations in performance of the model and we get away with a much smaller $\beta = 5$.

The results reported for the LIDC-IDRI dataset in Table 1 are based on the model configuration (including hyperparameters such as learning rate and batch size) obtained for the PCam dataset. This further strengthens the case for Tensor Network based methods as they can be easily transferred to different datasets.

In Table 1, we also report the GPU memory requirement for each of the models. Tensor network models require only a fraction of the memory utilised by the corresponding DenseNet or Rotation Eq-CNN models even when the number of parameters in LoTeNet is higher ($1M$ when compared to $120,000$ for the other two models (Veeling et al., 2018)). This drastic reduction in GPU memory utilisation is because tensor networks do not maintain massive intermediate feature maps, unlike CNNs which use a large chunk of GPU memory mainly to store intermediate feature maps (Rhu et al., 2016). As the entire pipeline of LoTeNet is based on contracting input data into smaller tensors it does not grow in memory consumption with successive contracted layers. This can be an important feature in medical imaging applications as larger images and larger batch sizes can be processed.

In conclusion, the proposed model, LoTeNet, overcomes the loss of global structure due to flattening in tensor networks using locally orderless regions that are added to the feature dimension of the input image. By using a hierarchical approach, the model also retains the global structure. We have demonstrated the ability of the model to perform classification on two publicly available datasets, yielding performance comparable to state-of-the-art deep learning models – using fewer model hyperparameters and substantially smaller GPU memory consumption.

## Acknowledgements

The authors would like to thank Silas Ørting and Mathias Perslev for their useful feedback on the manuscript. The authors also thank the four anonymous reviewers and the Area Chair for their insightful feedback which has strengthened the manuscript enormously.

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

## Appendix A. Supplementary Material

### A.1. Further details on LIDC Dataset

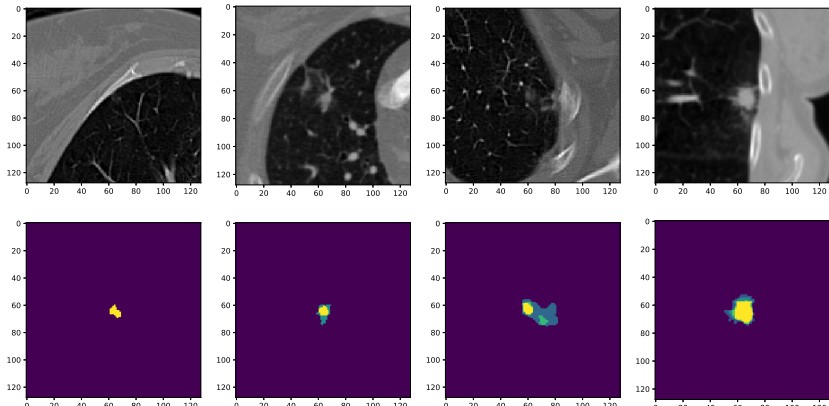

Figure 6: Four instances where the different raters do not agree. From left to right: One rater, two raters, three raters and four raters, indicated presence of nodules. In the binary task formulation, the first two patches will have a negative label and the last two will have positive labels.

### A.2. Model selection using PCam dataset

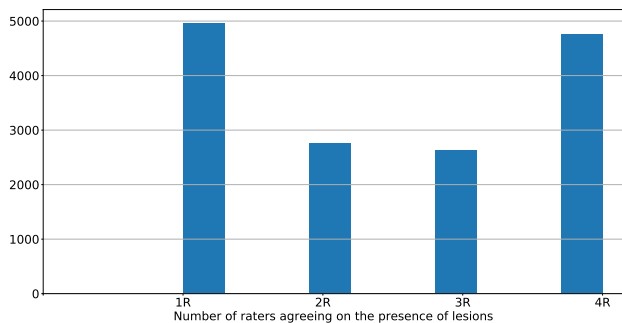

Figure 7: Histogram of number of raters agreeing on the presence of a lesion in each image. We merge the 1R and 2R classes to form the negative class and merge 3R and 4R class to obtain the positive class. It naturally leads to a well balanced data set.

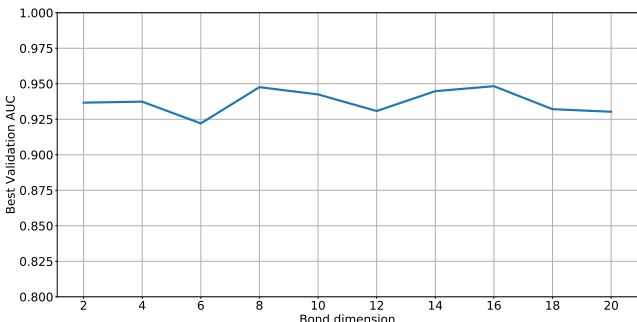

Figure 8: Influence of varying the bond dimensions reported with the best validation AUC on the PCam dataset.

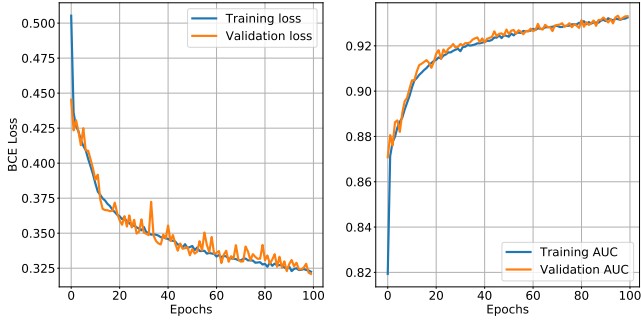

Figure 9: Learning curve for our model showing the evolution of the loss and AUC for training and validation data.

