# OpenReview forum: "Tensor Networks for Medical Image Classification"
_MIDL.io/2020/Conference — MIDL 2020_

### Official Review · AnonReviewer3 · 2020-03-10
**Very interesting and novel at first sight, but many questions arise when reading in detail**

**Rating:** 1
**Confidence:** 3
**Recommendation:** Poster

**Summary:**

The paper proposes an interesting idea to apply Tensor Networks (Stoudenmire-Schwab, Neurips 2016) to medical image data. The paper reads well and gives a nice explanation of tensor networks how to apply them to image data. The goal of using tensor networks is to learn feature embeddings of high dimensional data with a small(er) number of parameters, the latter is achieved via tensor decompositions. Results show that good performance can be achieved with much less parameters that the state-of-the-art. However, reading the paper raises several questions regarding the soundness of the approach.

**Strengths:**

The paper presents a new idea for analysis of (high)-dimensional medical image data via tensor networks. Parameterizing neural networks via tensor networks may be a good way to reduce the parameter search space and come to efficient architectures.

**Weaknesses:**

The premise of the paper is that the dimensionality of an image with N pixels and d channels is represented by a vector of dimension d^N, which would mean it consists of 2^10000 scalar values for an image of size 100x100 with 2 channels, this should be N*d instead… The rest of the paper builds on this analysis which I believe is not correct, and shines serious doubts on the usefulness/soundness of the approach.

What I furthermore feel is missing is a discussion on how the method relates to CNNs. The method (via tensor nets) in itself is not equivariant (important property in medical image analysis), but the authors design the architecture in such a way (via patch-wise processing) that it keeps the structure of the images mostly intact. In essence the method seems to describe a form of strided convolutions with convolution kernels parameterized via a tensor decomposition (see detailed comments).

The motivation for the paper is to reduce computational recourses and parameters, but the experiments are not setup to draw strong conclusions with respect to this. Moreover, the patch sizes are only of size 4x4, which isn't very large at all. Although the experiments show that indeed good performance can be achieved with the proposed framework with a small number of parameters, it is unclear if this couldn’t also be achieved with regular CNNs with fewer parameters, especially considering the similarities of the proposed work to regular CNNs (see discussion below w.r.t. strided convs). So, a good control/baseline is missing.


**Detailed Comments:**

[A: Dimensionality of W]

After Eq. (2) it is stated that the dimensionality of phi(x) is d^N, but this should be d*N. The image consists of N pixels, each pixel is mapped to vector of length 2. The indexing also does not seem to be right in this equation, is this perhaps the source of confusion? I believe this d^N comes from many particle physical (quantum) systems (which inspired MPS) where each particle can be in d states. The entire state space then consists of d^N possible configurations, you could call it the size of the space, but not the size/dimension of one instance/vector in this space.

Then the computation of dense parameterization of W^m is also incorrect. If phi(x) consists of N*d elements, and W^m maps to a vector of size M then W^m consists of at most N*d*M elements, and not M*d^N, which would indeed be incredibly large…

[B: Does the paper describe strided convolutions with factorized kernels?]

I believe the answer is yes but encourage the authors to identify flaws in my reasoning, which is as follows. The paper primarily revolves around the linear model W^m which turns a vector (image) into a prediction (or class scoring). W^m phi(x) really is just an inner product of a weight vector W^m with image/feature map phi(x)*, or conversely, W phi(x), describes a fully connected layer from image (patch) phi(x) to an M-dimensional output vector. Convolutions are nothing more than moving such a linear mapping around, turning local image patches into feature vectors for patches centered at every position in the image. When stride equals the kernel size, there is no overlap between patches, just like in this paper (Fig 4) where the image is divided in non-overlapping patching and each patch is transformed via a linear mapping W. So when, W is shared over the patches (not specified in the paper), the proposed method is a convolution with stride equal to the patch size and which the convolution kernel (W) is factorized via a tensor decomposition.

*Phi is a point-wise transformation. The given phi in Eq. (2) e.g. turns a single channel image into a two-channel feature map via cos/sin transform. It is a predefined embedding (essentially up to the user to define) that is not optimized over so we can omit this from our analysis.

I find the convolution discussion important for the following reason. Image data is highly structured, and to make use of this structure, and preserve it, the mapping W should be equivariant to translation. This is precisely why convolution layers are so successful in medical image analysis/computer vision, and fully connected layers are typically avoided. This paper proposes an approach that, at first sight, pays no attention to this structure and just aims to learn feature embedding via a (fully connected) linear mapping W. However, luckily in section 3 it turns out the image structure is somehow considered via particular squeeze operators and patch-wise application, which, if my analysis is correct, leads to a very complicated formulation of strided convolutions.

The strength of this paper then seem to lie in the fact that large (?) convolution kernels can be efficiently parameterized via a tensor decomposition. However, in the end patches are only of size 4x4 (k=2) in the PCam case, which would correspond to 4x4 convolution kernels with stride 4.

With this in mind, I am left wondering if the same results could also be obtained in a very simple convolutional architecture with large convolution kernels and stride (equal to patch size).

[C: Where are the non-linearities?]

Tensor products are linear. A whole chain (Fig 4) of tensor contractions is also linear, and may as well be described in one big matrix vector multiplication. Are there details missing in the paper? Or does the model really describe only a linear projection?

[D: Extreme over-parameterization in Eq (5)?]

The whole structure seems to be more computationally demanding that necessary. E.g. Eq. (5) contracts over a lot of indices to construct a W^m which is really only of size 4x4 (for k=2), and the contraction dimensions is beta=5, which means each A^{i_j} is of size beta*beta and the algorithm takes N beta^3 d = 16*125*5=10.000 multiplications (d=beta). Whereas in the regular setting with W^m be of length 16*d, you would only need 16*5= 80 multiplications. Isn’t W^m way over-parameterized via Eq 5?

[E: Small comment]

When keeping track of the input size at each layer, given Figure 4, there’s a discrepancy with the PCam patch size which reduce with L=3 and k=2 via: 96/4->24 /4 -> 6 /4 -> 1?


**Justification Of Rating:**

I recommend reject based on the concerns that I express in this review. I hope I am mistaken in my analysis and overlooked something, but with my current understanding I cannot accept the paper as I think there are some major flaws in the paper (mainly on the dimensionality of W and the need for tensor networks). I would encourage the authors to use the rebuttal period to alleviate my concerns, if possible, and update the paper accordingly.

**Paper Type:**

methodological development

**Questions To Address In The Rebuttal:**

My feeling is that the premise of the paper is fundamentally wrong. See detailed comments ([A: Dimensionality]. In case I am incorrect, could you please indicate the error in my reasoning?

A comparison with a CNN baseline with about the same network capacity is required to say something about the need for tensor networks. Could you provide these?

Does the paper in essence describe strided convolutions with factored kernels? (see detailed comments)

Could you comment on the use of non-linearities in the architecture? (See detailed comments)

Are the layers not heavily over-over-parameterized? (see detailed comments)

**Special Issue:**

no

---

> ### Author Response · Authors · 2020-03-27
> **Response to Reviewer 3**
>
> We thank the reviewer for their interest in our work and the detailed comments. In our opinion, some of the major concerns are due to ambiguities in the notations arising from introducing several new concepts related to Tensor Networks in the paper.
>
> We address the primary concerns below:
>
> A & D. Dimensionality and Over-parameterisation:
>
> The ambiguity about dimensionality perhaps stems from the fact that there is not sufficient elaboration in Eq.(1) about the tensor product. We do mention that the joint feature map is obtained by taking tensor product of the local feature maps to obtain the high dimensional phi(x) using the commonly used tensor product notation (x). We illustrate this with a toy example below.
>
> Consider a vector of size N=3, x=[x1,x2,x3]. According to Eq.(2) we obtain the local feature map transformations with d=2 as,
>
> phi(x1) = [sin(x1),cos(x1)]
> phi(x2) = [sin(x2),cos(x2)]
> phi(x3) = [sin(x3),cos(x3)]
> Each of these local feature maps are in R^2.
>
> The joint feature map phi(x) in Eq.(1) is obtained by taking a tensor product of individual phi(xi) which results in a new vector space of dimensionality equal to the product of the local feature maps due to the property of tensor products  i.e,
> phi(x) = phi(x1) (x) phi(x2) (x) phi(x3) = phi(x1x2) (x) phi(x3),
>
> where phi(x1x2) is obtained by taking the matrix outer product (or tensor product) yielding a new tensor which is R^{2x2} = R^4. Continuing with another tensor product with phi(x3) which is in R^2 results in the "global" phi(x) which is R^{2x2x2} = R^{8}, which has 8 = 2^3 = d^N elements and not d*N = 2*3 = 6 elements.
>
> If we assume a binary classification task, we can take the inner product according to Eq.(4) with m=1, to obtain a single scalar output. Due to this, the weight matrix is also of the same dimension d^N so that the dot product (W,phi(x)) yields a scalar. This computation without MPS approximations would indeed be in a very high dimensional (and infeasible) space. The MPS algorithm is specifically used to approximate this inner product with d*N*beta parameters efficiently with O(Nd beta^3) [1] --- without explicitly having to deal with the exponentially large space.
>
> B: Strided convolutions
>
> The use of MPS block per patch can be interpreted as a form of strided convolution but only in how the fully connected operation is performed at each patch. The primary difference of using MPS blocks in LoTeNet compared to strided convolutions is in weight sharing; the weights of MPS blocks are not shared across the image. Each region (kxk patch) has a local MPS block acting on it. This is indicated in the description of the model and in Figure 4, where we report the number of MPS blocks used per layer. We will clarify this further to emphasize that there is no weight sharing of MPS blocks at each layer.
>
> We agree that image data is highly structured and that CNNs exploit this feature; this is also taken into account in LoTeNet albeit in a different manner as recognized by the reviewer. One of the primary contributions of adapting Tensor Networks for 2d images is retaining the global image structure. This is achieved by operating at different resolutions with locally orderless operations performed with MPS blocks. In locally orderless image analysis, statistics from local regions are aggregated at different resolutions to summarize the image structure. The LoTeNet model takes inspiration from locally orderlessness analysis and uses MPS blocks to extract local statistics from high dimensional spaces according to Eq.(4). With the hierarchical structure we are able to  retain global image information which is the primary difference when compared to the method in [3].
>
> C: Non-linearities
>
> There are, in fact, no non-linearities except in the last layer which uses a soft-max operation to obtain class scores. This is in line with the hypothesis that we are seeking a linear decision boundary in a sufficiently high dimensional space to perform the classification task.
>
> E: Small Comment
>
> "with L=3 and k=2 via: 96/4->24 /4 -> 6 /4 -> 1"
> Here, the reviewer assumes N=96. According to the notations introduced in Section 2.2, N is the total number of pixels i.e N=96x96.
>
> The transformation according to Fig. 4 will be:
>
> First layer: 96x96/4 MPS blocks
> Second layer: 48x48/4 MPS blocks
> Third layer: 24x24/4 MPS blocks
> And a final layer with 12x12 MPS blocks as mentioned in Sec. 3.1.
>
> Additional CNN based experiments:
>
> Other than the DenseNet baseline we report experiments with Rotation Equivariant CNN [2] which is the state-of-the-art for PCam classification task, and LoTeNet model attains close to this method as reported in Table 1.
>
> [1] Matrix Product State Representations, D. Garcia, et al (2007)
> [2] Veeling, Bastiaan S., et al. "Rotation equivariant CNNs for digital pathology." International Conference on Medical image computing and computer-assisted intervention. Springer, Cham, 2018.
> [3] Efthymiou et al "TensorNetwork for machine learning." (2019).

---

> > ### Comment · AnonReviewer3 · 2020-03-28
> > **Thank you for the additional explanation**
> >
> > I thank the authors for their detailed responses, this much appreciated. Regarding my main concern (A): I stand corrected and thank the authors for their explanation. This also explains to me the difference with strided convolutions. My apologies for the misunderstanding, this should have been clear to me from the start. A small extra clarification in the text would nevertheless be appreciated.
> >
> > Although this convinces me that the paper is indeed sound, I still think a more thorough analysis of tensor net architectures vs CNNs is appropriate, see also comments of other reviewers. Also, the broader context, e.g., the relation/differences to CNNs could be better explained.
> >
> > B
> > Regarding the patch-wise processing. It is explained that there is no weight sharing among patches. Why are the weights not shared? A tumor cell should look the same whether it be in the upper left corner patch, or at the center of the image. As such, I see no reason why each block (within the same hierarchy) is allowed to have its own set of weights. I believe the locally orderless concept does not prevent networks to be biased towards spatial regions/patches, if I am right (which I dare not to assume again :p).
> >
> > C
> > I now understand that not using non-linearities makes sense, considering the huge dimensionality of the initial representation.

---

> > > ### Comment · AnonReviewer3 · 2020-03-30
> > > **Change of score to "3 weak accept"**
> > >
> > > In light of the rebuttal I'd like to change my score from strong reject to "3 weak accept".

---

### Official Review · AnonReviewer4 · 2020-03-15
**an interesting application of tensor network on medical images.**

**Rating:** 3
**Confidence:** 4
**Recommendation:** Poster

**Summary:**

Model parameter reduction has a very crucial need in processing medical images. Tensor networks have achieved a great success in reducing parameters in other machine learning tasks. This paper applies the tensor network method to medical image classification tasks. Different from [1], this paper adopts the local orderlessness concept and designs a multi-layer structure. This adaptation leads to fewer parameters and higher performance in image classification tasks of PCam and LIDC.
[1]Efthymiou S, Hidary J, Leichenauer S. TensorNetwork for machine learning[J]. arXiv preprint arXiv:1906.06329, 2019.

**Strengths:**

The motivation of applying tensor networks for model parameter reduction in processing medical images is good.  Experimental results have demonstrated the good performance. Overall, this paper is well-organized and well-written.

**Weaknesses:**

1. Detailed discussion on the connection to [1] is needed.
2.  A table of comparing space complexity could be provided.
3. How to reshape output vectors for a given layer back into an image could be explained more, since the output of (Eq.4) does not satisfy the quantum wave property of (Eq.2) directly.
4. This paper provides only the time cost of LoTeNet. The time cost of other models could also be provided.


**Detailed Comments:**

See above.

**Justification Of Rating:**

The idea of applying tensor networks for reducing parameters in medical images is a good try for medical images. The results look good, however, complexity analysis  and running time comparison is important to demonstrate the advantages of this paper.

**Paper Type:**

validation/application paper

**Questions To Address In The Rebuttal:**

1. Detailed discussion on the connection to [1] is needed.
2.  A table of comparing space complexity could be provided.
3. How to reshape output vectors for a given layer back into an image could be explained more, since the output of (Eq.4) does not satisfy the quantum wave property of (Eq.2) directly.
4. This paper provides only the time cost of LoTeNet. The time cost of other models could also be provided.

**Special Issue:**

no

---

> ### Author Response · Authors · 2020-03-27
> **Response to Reviewer 4**
>
> We thank the reviewer for their constructive feedback and recognizing the contribution of extending Tensor Networks to medical imaging tasks. We address the reviewer’s concerns below.
>
> Connection to [1]
> The primary contribution in this work is to extend tensor networks, specifically the MPS approximation/ tensor train networks, to large 2d images that are encountered in medical imaging. This was achieved by approaching the task using locally orderless analysis resulting in the hierarchical use of MPS blocks on local regions of the images.
> This model differs from [1] in the following regards:
> -- Hierarchical modelling: The method in [1] flattens 2d images into 1d vectors and learns a linear decision boundary in a high dimensional space using MPS approximations. Flattening 2d images into 1d vectors results in loss of useful global structure of images. In LoTeNet, we overcome this loss of structure by using layers of MPS blocks in a hierarchical manner. At each layer, small regions of the image are embedded using MPS blocks to obtain an embedded representation. The MPS acting on each of these patches is similar to the model used in [1].
> -- Local feature map: A smaller difference is the use of sinusoidal feature maps in Eq.(1) instead of the [p,1-p] local feature transformation used in [1], where p is the pixel intensity.
> Space complexity
> We are already reporting the number of parameters used in each of the models in Section 5 as a proxy for the space complexity. We have not conducted a comprehensive analysis of the space complexity of the baseline models. We base our crude bounds on [2]. We will update these to include the space complexity which is reported below:
>
> DenseNet:              $\mathcal{O}(2^{h \cdot N})$ [2]
> Tensor Net-X[1]:    $\mathcal{O}(N\cdot d\cdot \beta^2)$
> LoTeNet:                 $\mathcal{O}((N/(k^2))\cdot L \cdot d \cdot \beta^2)$ [3]
>
> where we use the big-O notation to indicate an upper bound, h is the number of hidden layers in CNN architectures, N is the number of pixels per input image, d is the dimensionality of feature space, $\beta$ is the bond dimension, k is the kernel stride of squeeze operation, and L is the number of MPS layers.
>
>
> Reshaping intermediate layer outputs:
> As the reviewer points out, the output of Eq. 4 is not in image space. For each patch, the MPS block outputs a 1d vector of length equal to the virtual dimension, v.
> And as indicated in Fig. 4, each layer outputs $N/(k^{2*(l+1)}) $ vectors of length v. These are then reshaped back into the (reduced) image space using a standard reshape operation before squeezing them into the next layer.
> Concretely,
> At First layer: 96x96/4 =2304, these are reshaped back to 48x48 in the image space, and so on.
> In the current version of the paper, this is shown in the code excerpt as the “Unravel to 2D image space” operation.
>
>
> Computation costs:
> The numbers on the PCam dataset per training epoch are reported below.
> LoTeNet    185s
> DenseNet    120s
> Rot. Equiv.    175s
> Tensor NetX    160s
> The tensor network computation time can be further reduced by developing specialized tensor contraction modules. By now most of the CNN operations are highly optimized whereas the Tensor Network packages [4] suffer from lack of specialized operations primarily in performing the MPS approximations.
>
>
> [1] Efthymiou, Stavros, Jack Hidary, and Stefan Leichenauer. "TensorNetwork for machine learning." arXiv preprint arXiv:1906.06329 (2019).
> [2] Bianchini, Monica, and Franco Scarselli. "On the complexity of shallow and deep neural network classifiers." ESANN. 2014.
> [3] Stoudenmire, Edwin, and David J. Schwab. "Supervised learning with tensor networks." Advances in Neural Information Processing Systems. 2016.
> [4] Jacob Miller. Torchmps. https://github.com/jemisjoky/torchmps, 2019.

---

### Official Review · AnonReviewer2 · 2020-03-16
**An interesting use of Matrix Product State via Tensor Network to deal with large images**

**Rating:** 3
**Confidence:** 3
**Recommendation:** Poster

**Summary:**

A tensor network which models  Matrix Product State (MPS) is presented. This is an efficient approximation of a naive tensor representation which has been used in related fields. The performance is tested on a couple of public datasets: binary classification of metastasis and CT lesion detection. The presented method showed competitive AUROC results with significantly smaller model sizes.

**Strengths:**

1. Despite the technicality, it is overall nicely written.
2. The reduction of model parameters is a significant gain and has potential practicalities in large images.
3. The ablation studies of the parameters (e.g., bond dimension) are helpful.

**Weaknesses:**

1. The squeeze operation is not particularly novel or interesting. Real-NVP was using it for the purpose of splitting the input feature into two partitions (by construction of model), so I am not sure if the intentions are the same.

2. AUROC is the only metric.

3. Please see the "Questions To Address In The Rebuttal" section.

**Justification Of Rating:**

The overall paper is interesting, although the technical novelty is resembling several related works in the field that I have mentioned. There are several questions I would like to hear back from the authors, and I am willing to raise my score based on the response.

**Paper Type:**

methodological development

**Questions To Address In The Rebuttal:**

My impression is that the Tensor Network presented in this paper is quite similar to the Tensor Train Network (Tensorizing Neural Networks by Novikov et al., 2015) which is based on Tensor Train Decomposition (Oseledets, 2011) which actually characterizes the MPS. In fact, Exponential Machine (Novikov et al.) was mentioned in Section 2.2. I am curious if there are differences or similarities between the Tensor Train approaches and the presented LoTeNet.

Similarly, I wonder how similar or different the squeeze operation is to the reshaping/indexing of matrix introduced in Tensorizing Neural Networks (Novikov et al., 2015).

Lastly, would the model be able to perform squeeze operation on non-square images?

**Special Issue:**

no

---

> ### Author Response · Authors · 2020-03-27
> **Response to Reviewer 2**
>
> We thank the reviewer for their interest in our work and their constructive feedback. We address the major concerns below and will make the necessary changes to the manuscript.
>
> Connection to Tensor Train Networks [1,2]
> We thank the reviewer for pointing out the specific references [1,2] which were missed in our submission. We will include a discussion to contextualize our contribution with respect to the tensor train network approaches.
> The MPS blocks used in this work are in fact identical to the tensor train networks [2]. Recently, the machine learning community is adopting the MPS nomenclature more than tensor train networks and as [3,4]  are our primary related works we inadvertently have continued this trend. This will be corrected to clarify that MPS and tensor train network approaches are identical with due reference to [1].
>
> Comparison with Exponential Machines [5]
> The single layered MPS method for image classification proposed in [4] which we compare to, reported as Tensor Net-X in Table 1, is similar to the baseline model used in [5] which utilizes stochastic gradient descent to optimize the parameters of the tensor train cores. The proposed LoTeNet differs from Exponential machines primarily in these regards:
>
> -- Hierarchical modelling of the image data to capture global image structure which is essential in images encountered in medical imaging.
> -- There is no embedding of pixel intensities with local feature transformation in Eq.(1) and Eq. (2) to construct a further higher dimensional input representation in Exponential Machines. This transformation step is essential in LoTeNet.
> -- The optimization we use is similar to [5] based on the Adam optimizer unlike the Reimannian optimization used in Exponential machines. This is less of a methodological difference and more of a choice of optimization strategy. We expect that LoTeNet can be optimized using the strategy used in Exponential Machines.
>
>
> Squeeze operation
> The squeeze operation is not a substantial methodological contribution in LoTeNet; however, it is an essential component of the architecture. This is indeed a sequence of reshape operations and we connect this to the well known squeeze operation in Real NVP [6]. The purpose of squeeze operations in Real NVP and LoTeNet are also quite similar. We are also interested in reducing spatial resolution without losing useful information. We stack the spatial information into the feature dimension while reducing the spatial dimension.
>
>
> Non-square images
> In the current formulation, we assume the images are rectangular. The simplest strategy to work with non-square (circular images for instance) would be to pad the regions around the borders to make them rectangular. This might not be a critical issue as we use small kernel sizes to perform the squeeze operation.
> The purpose of the squeeze operation is to reduce the spatial dimension and increase the feature dimensions. This can be achieved by reshaping any small, non-square region of the image into a vector of fixed feature dimension. For instance, a non-square image can be partitioned using Voronoi cells and each of those cells can be transformed into a 1d vector.
>
> [1] Novikov, Alexander, et al. "Tensorizing neural networks." Advances in neural information processing systems. 2015.
> [2] Oseledets, Ivan V. "Tensor-train decomposition." SIAM Journal on Scientific Computing 33.5 (2011): 2295-2317.
> [3] Stoudenmire, Edwin, and David J. Schwab. "Supervised learning with tensor networks." Advances in Neural Information Processing Systems. 2016.
> [4] Efthymiou, Stavros, Jack Hidary, and Stefan Leichenauer. "TensorNetwork for machine learning." arXiv preprint arXiv:1906.06329 (2019).
> [5] Novikov, Alexander, Mikhail Trofimov, and Ivan Oseledets. "Exponential machines." arXiv preprint arXiv:1605.03795 (2016).
> [6] Dinh, Laurent, Jascha Sohl-Dickstein, and Samy Bengio. "Density estimation using Real NVP." (2016).

---

### Official Review · AnonReviewer1 · 2020-03-20
**Tensor networks for image classification**

**Rating:** 3
**Confidence:** 4
**Recommendation:** Poster

**Summary:**

The paper investigates tensor networks for medical image classification, specifically the use of Matrix Product State (MPS) blocks as an alternative to standard convolutional architectures. In this context, MPS blocks embed input multichannel patches into a vector representation, and so on for every layer until the final output is plugged into a classification (softmax) layer.

The use of tensor network is to my knowledge not mainstream in the community. The main technical contribution of the paper seems to be to adapt the framework for 2D images, where capturing the local and global structure is important.

The approach is illustrated on two datasets / classification tasks, PCam (presence of tumour tissue) and LIDC (presence of lesion). The proposed LoTeNet is compared to a 1-layer MPS architecture, and to a DenseNet architecture.

**Strengths:**

- The framework investigated here is not mainstream
- The paper is overall well-written and well-structured, although it is hard to follow at times (around Eq. 5, the wording is confusing, when introducing $\alpha$ and $\beta$; if I understood correctly the word "dimension" is used for several clashing purposes here.)
- Overall it is intriguing as an alternative approach to pattern detection / non-linear embedding and I could see other applications for some of the core ideas.


**Weaknesses:**

Validation is a bit limited. The main advantages of the approach can also be its main drawbacks.

For instance the reduction in GPU memory footprint seems to be due to directly embedding patches, rather than computing full feature maps then pooling. This also means that the approach will be more difficult to extend outside of classification, or to more complex classification architectures (not purely feedforward). Also, wouldn't convolutional architectures computed with a stride equal to the kernel size benefit from the same improved footprint?

Regarding accuracy, it would be useful to clarify the impact of the number of parameters compared to benchmark architectures. "the number of parameters in LoTeNet is higher (1M when compared to 120,000 for the other two models)"

**Detailed Comments:**

Regarding clarity, I would suggest disambiguating the use of "rank" and "dimension", maybe by using "order" or "dimension" as appropriate (I'm not sure you ever mean the actual rank, except maybe with the $\alpha_j$'s in Eq. 5). It should help the reader when $\alpha$ and the bond dimension are introduced.

**Justification Of Rating:**

The paper is, on the overall, well structured; the approach has some originality and the work is well-suited for MIDL. On the other hand, the validation is a bit limited so that it is difficult to truly judge the actual usefulness/benefit of using tensor networks for medical image classification. What misses most is some insight into how the MPS block works (not necessarily the 2D adaptation which is well-illustrated; but rather what Eq. 1,2 and 5 concretely result in), compared to convolutions or other embeddings, and the reader has to invest a bit of time figuring it out for themselves.

**Paper Type:**

methodological development

**Special Issue:**

no

---

> ### Author Response · Authors · 2020-03-27
> **Response to Reviewer 1**
>
> We thank the reviewer for their insightful feedback on the paper that identifies the contribution in adapting tensor networks for 2d medical images. We address the reviewer’s concerns below.
>
> Limited Validation:
> We compare the performance of LoTeNet on two datasets against two different CNN architectures and the single layer MPS method. In all cases, the performance and the corresponding reduction in GPU memory utilisation is consistent.
>
> Difference to strided convolutions:
> The use of MPS block per patch can be interpreted as a form of strided convolution but only in how the fully connected operation is performed at each patch. The primary difference of using MPS blocks in LoTeNet compared to strided convolutions is in weight sharing; the weights of MPS blocks are not shared across the image. As the reviewer points out, the reduction in GPU memory is due to the embedding of patches in LoTeNet. Even when using strided convolution layers, this extent of reduction in GPU memory might not be possible due to the computation of intermediate feature maps, as the bulk of memory utilization is due to the feature maps [2] as discussed in Section 5.
>
> Extension to more complex tasks:
> While neural networks can be seen as learning non-linear decision boundaries in low dimensional spaces, tensor network based models learn linear decision boundaries in very high dimensional spaces. This fundamental difference between the two classes of methods might give them different advantages. In this work, we focused on extending tensor networks to 2d medical image classification and have shown they can be competitive with relatively small GPU memory requirements. Concretely, this comparison was demonstrated by comparing the LoTeNet model with the Rotation Equivariant CNN model [1] which is the state-of-the-art model on the PCAM dataset, where we attain reasonably close performance. However, it is still unclear as to how one could extend tensor networks to semantic segmentation tasks where fully convolutional networks are state-of-the-art which is a research direction we will be pursuing.
>
> Number of parameters:
> Due to the non-weight sharing aspect of the MPS blocks the number of parameters in LoTeNet is potentially higher than in the CNN based models. In our experiments we did not see this to be an issue both in computation time and overfitting issues. However, when dealing with small datasets, which are more common in medical imaging, Tensor Network models might be more prone to overfitting issues. This can be alleviated by reducing the bond dimension which controls the number of parameters. For instance, bond dimension of 2 reduces the number of parameters from 1M to about 200K. As illustrated in Fig 8, there is no reduction in performance for smaller bond dimensions in our experiments.
>
> Rank and dimension: Our aim was to connect the tensor network concepts to other common linear algebra ideas. But we now recognize this has led to confusion. We will as pointed out by the reviewer drop the mention of rank in tensor network introductions and adhere to describing them with dimensions which are more consistent with tensor network literature.
>
> [1] Veeling, Bastiaan S., et al. "Rotation equivariant CNNs for digital pathology." International Conference on Medical image computing and computer-assisted intervention. Springer, Cham, 2018.
> [2] Rhu, Minsoo, et al. "vDNN: Virtualized deep neural networks for scalable, memory-efficient neural network design." 2016 49th Annual IEEE/ACM International Symposium on Microarchitecture (MICRO). IEEE, 2016.

---

### Author Response · Authors · 2020-03-27
**Errata**

In Eq. (2), the subscript index of x on the left hand side has to be j and not i.
$\phi^{i_j}_{j} (x_j ) = [cos( \frac{\pi}{2} x_j ),sin(\frac{\pi}{2} x_j )]$.

---

### Meta-Review · Area_Chair1 · 2020-04-06
**MetaReview of Paper95 by AreaChair1**

**Rating:** 4
**Recommendation For Accepted Papers:** Oral

**Metareview:**

This submission explores a very interesting concept, implicitly modelling high-dimensional decision boundaries, that has not been used a lot in medical deep learning. They build upon related work from computer vision and extend the concept to larger images, capturing both local and global information. All reviewers show high interest in this work and recommend acceptance (at least after the rebuttal). There was a fruitful discussion among reviewers and authors that will subsequently improve the final version. While the experimental validation is limited to rather small 2D patches (128x128) the results are promising and there are may certainly come interesting future papers that further extend these concepts.

**Paper Type:**

methodological development

**Special Issue:**

no

---

### Decision · Program_Chairs · 2020-04-11

Accept